# A Shift in Asthma Treatment According to New Guidelines: An Evaluation of Asthma Patients’ Attitudes towards Treatment Change

**DOI:** 10.3390/ijerph20043453

**Published:** 2023-02-16

**Authors:** Sara Sommer Holst, Ebru Sabedin, Esin Sabedin, Charlotte Vermehren

**Affiliations:** 1Department of Clinical Pharmacology, Copenhagen University Hospital Bispebjerg, 2400 Copenhagen, Denmark; 2Department of Drug Design and Pharmacology, Faculty of Health and Medical Sciences, University of Copenhagen, 2200 Copenhagen, Denmark

**Keywords:** asthma, pharmacological asthma treatment, maintenance and reliever, treatment change, treatment shift, attitude, barriers, facilitators, supportive initiatives

## Abstract

The Global Initiative for Asthma (GINA) has presented a shift in pharmacological asthma treatment. The objective of this study was to explore factors influencing a successful switch to a new asthma treatment approach with a focus on asthma patients’ attitudes toward treatment change and supportive initiatives. This study was performed as a case study involving a quantitative questionnaire and a qualitative semi-structured interview. A total of 284 responses were collected from the questionnaire, and 141 responses were included. The results showed that asthma patients thought that effectiveness of the new treatment approach, doctor recommendation, and knowledge of the new treatment approach were the most important factors influencing treatment change considerations. Nine interviews were conducted where the main themes were barriers to a shift in asthma treatment, such as effects and side effects of the new treatment, the role of the general practitioner (GP) and conflicts in agreeing on a treatment plan; as well as facilitators to a shift in asthma treatment, such as trust in the GP and easier inhaler use. We found several supportive initiatives, such as consultation with the GP, handing out information leaflets and a consultation at the pharmacy. In conclusion, this study uniquely identified factors that may influence successful treatment shifts in asthma patients that may be instrumental in understanding similar situations in other pharmacological settings.

## 1. Introduction

Asthma is a common disorder with a prevalence of 7–11% in Denmark and is characterized by chronic respiratory inflammation [1]. The incidence is highest in childhood and adolescence, but the disease can occur at any age. The disease is often characterized by respiratory symptoms, such as wheezing, dyspnoea, chest tightness and coughing, that can vary over time and in intensity [1].

The pharmacological treatment of asthma depends on the severity of the disease, where the aim is to make the patient asymptomatic, achieve normal or best possible lung function and reduce the risk of acute flare-ups and exacerbations [2]. Patients with asthma will therefore receive both maintenance and reliever therapy, depending on the severity.

The Global Initiative for Asthma (GINA) presents a global strategy for asthma management and prevention. The strategy is updated yearly. In 2019, new GINA recommendations were made for the pharmacological treatment of adult asthma patients [3]. These recommendations have been slightly updated in the newest GINA guidelines from 2022 [2]. In Denmark, the GINA recommendations are presented by the Danish Respiratory Society in a national guideline for the treatment of mild to moderate asthma. However, due to the COVID-19 epidemic, the national guidelines were not updated in 2020, which meant that the international recommendations from GINA 2019 were only implemented in the Danish national guidelines in 2021 [4].

In the new guidelines from GINA, the preferred line of treatment, which includes all asthma patients regardless of their severity, recommends inhaled corticosteroids (ICS) in combination with a long-acting β_2_-agonist (LABA) as reliever therapy. Based on asthma severity, this treatment can also be used as maintenance therapy (MART) [2,3]. When choosing a LABA in the combination treatment, GINA recommends the use of formoterol due to its long and rapid onset of action [2]. A meta-analysis by O’Shea et al. from 2021 updated the evidence on the safety profile and risk assessment of the combination of formoterol and ICS versus the combination of salmeterol and ICS, and found no evidence of safety issues that could affect the choice between salmeterol and formoterol in combination inhalers used for regular maintenance therapy by adults and children with asthma [5]. Finally, GINA does not recommend asthma patients being treated with short-acting β_2_-agonists (SABAs) in monotherapy [2].

A summary of the GINA Guidelines from 2018 compared to the GINA guidelines from 2022 is shown in Appendix B.

The MART approach ensures the daily coverage of ICS while providing additional doses of ICS as soon as symptoms appear. This results in a reduction of 40–50% of exacerbations compared with using ICS–formoterol as a preventive treatment as well as a short-acting β_2_-agonist (SABA), as needed [2]. Even patients with mild asthma can reduce the risk of severe exacerbations and hospitalization if treated with ICS due to the anti-inflammatory effect [6,7].

The new MART approach presents many advantages but also requires that the patients themselves can assess their asthma symptoms to a greater extent and respond to them by administering their inhaled medication, as needed, as fewer patients will receive fixed daily dose prescriptions. However, poor adherence to asthma treatment is already a known and common problem, ranging from 30–70% among asthma patients [8]. Nonadherence is associated with poor health care outcomes, such as inadequately controlled asthma and an increased risk of exacerbations [8,9,10,11]. Therefore, a continuous focus on improving the adherence of asthma patients is crucial, especially when implementing a new treatment approach.

Studies have investigated adherence and how to improve adherence in asthma patients, where a lack of self-management skills, poor health literacy knowledge and lack of support from health care professionals were important factors influencing adherence [12,13,14]. A new observational study has, amongst other things, investigated asthma patients’ treatment adherence and whether there were any differences between conventional fixed combined therapy and the new MART approach after inhaler education. The study found that asthma treatment adherence improved equally with inhaler education in both groups [15]. However, there is a lack of knowledge regarding adherence in asthma patients when they are changing their asthma treatment. Thus, the objective of this study—which, to our knowledge, has not previously been investigated—was to identify potential factors that may be crucial for a successful, i.e., adherent, shift to a new asthma treatment approach. The study specifically focused on the factors below:Asthma patients’ attitudes toward changing their treatment according to new treatment guidelines.How asthma patients can be supported in the decision of making this recommended treatment shift.

## 2. Materials and Methods

### 2.1. Design

The study was performed as a case study exploring how a group of Danish asthma patients would handle a specific situation: a switch to the new MART approach.

A case study design is a good way to explore and understand a defined setting, i.e., providing in-depth information about the study objective based on a smaller sample size [16]. The researchers of the present study had a social constructionist view on research, where meaning and knowledge are constructed by human beings as they interact and engage in interpretation [17].

It was assumed that all asthma patients were in a controlled asthma treatment, and no asthma patients were recommended to adjust their treatment according to the inclusion criteria. We purely investigated asthma patients’ attitudes toward a potential medication change from the conventional treatment approach to the new MART approach on a correlating asthma severity step.

In summary, this project aimed to identify factors which may be decisive for asthma patients to be willing to consider a treatment shift by exploring the patients’ attitudes toward a shift to the new MART approach. The case study involved two methods:A quantitative questionnaire method.A qualitative semi-structured individual interview method.

In the study, the individual respondent’s degree of asthma and associated medication were not assessed. The patients’ responses were evaluated solely based on their information about their asthma diagnosis, degree of asthma and inhalers used. However, an attempt was made by the authors to estimate and qualify the patients’ degree of asthma based on their information about used inhalers, in order to include only participants with mild to moderate asthma.

The study design and method description adhered to the JBI Critical Appraisal Checklist for Qualitative Research shown in Appendix A [18], and the Consolidated Criteria for Reporting Qualitative Research (COREQ) [19].

### 2.2. Inclusion and Exclusion Criteria

The inclusion and exclusion criteria were identical in both the quantitative questionnaires and the qualitative interviews. Inclusion criteria included asthma patients aged 18 and older who were eligible to change their medication. According to the target group of the MART approach, only patients suffering from a mild or moderate degree of asthma in controlled and stable pharmacological treatment were included. Exclusion criteria were patients suffering from chronic obstructive pulmonary disease (COPD), severe degree of asthma or treated with more than three inhalers, indicating severe asthma [2]. As described above, this distinction of asthma severity was determined by relating the participants’ own description of their use of prescribed asthma drugs to the correlating degree of asthma severity.

### 2.3. The Quantitative Questionnaire Method

Quantitative questionnaires were used to explore asthma patients’ knowledge of their disease and treatment as well as their attitude toward changing their treatment according to the new MART approach. The specific focus was to explore what information—e.g., pharmacological, economic, and practical instructions—asthma patients demanded before they could decide on the new treatment, and how asthma patients saw themselves best supported in the decision of making this treatment change.

The sample strategy was random selection, as the questionnaires were both distributed online in a Facebook group for asthma patients and face-to-face by staff at a local pharmacy, who handed out a hard copy version of the questionnaire. The questionnaires were identical, regardless of the distribution form.

The questionnaire was pilot-tested in 7 random persons, of which 4 had asthma, all with different sociodemographic characteristics.

### 2.4. The Qualitative Semi-Structured Individual Interview Method

Qualitative semi-structured individual interviews were used to explore asthma patients’ attitudes toward changing their treatment, according to the new MART approach, and to determine which barriers and facilitators may occur when changing. Furthermore, it was used to explore how asthma patients could be supported in such a process.

The interviews were performed at Bispebjerg Hospital and at the University of Copenhagen‘s School of Pharmaceutical Sciences.

The sampling strategy was a convenience strategy, as the patients were recruited from an opportunity provided in the questionnaires to state if they wanted to participate in an individual interview.

All interviews were conducted by pharmacists (EBS, ESS). The interview guide was pilot-tested on a third person. All interviews were audio-recorded, and field notes were made during and immediately after the interviews. The audio records of the interviews were transcribed verbatim. Transcripts were analyzed using systematic thematization [20]. Thematic coding analysis was used to analyze the individual interview data [17]. Themes were derived directly from the interview data and were not identified in advance. All participants were adequately represented, and examples of quotes that emphasized the viewpoints of the participants were highlighted.

## 3. Results

A total of 284 responses were collected in the period from 6 April 2022 to 27 April 2022, where 14 respondents were recruited at the local community pharmacy, and the rest were recruited from the online Facebook group for asthma patients. Interview participants were included from the questionnaire respondents. The inclusion and exclusion process is outlined in Figure 1.

### 3.1. Results of the Quantitative Questionnaire

Of the 284 respondents who completed the questionnaire, 142 were excluded (Figure 1). A total of 141 responses were included in this study. An overview of the questionnaire respondents is presented below in Table 1.

Respondents who used ICS and SABA in separate inhalers, SABA as monotherapy, the combination of ICS + LABA (not formoterol) in one inhaler, and ICS + LABA (not formoterol) and SABA met all requirements to potentially be considered for switching to the new MART approach if they had controlled and stable asthma in their current pharmacological treatment.

#### 3.1.1. Attitudes toward Asthma Treatment

To gain insight into the respondents’ knowledge of their pharmacological asthma treatment and thus their basis for making an informed decision about a possible shift in treatment, the respondents were questioned about their knowledge of the new MART approach. The vast majority (78%) of the respondents had never heard about the new approach. In addition, 31% of respondents had never sought information about their asthma treatment before nor read the patient information leaflet for their asthma medication.

Most of the respondents did not find it difficult to use their asthma inhaler correctly. However, one-third of the respondents (33%) agreed that it could be difficult to remember their asthma inhaler. Eighteen percent neither agreed nor disagreed with the statement, while the remaining respondents (49%) disagreed. Consequently, half of the respondents believed that their inhaled medication had a beneficial effect on their asthma, while 32% disagreed and 18% neither agreed nor disagreed. The vast majority (66%) of respondents believed that they needed their inhaled medication. The rest of the respondents (34%) did not feel a need for pharmacological asthma treatment as, according to the respondents, their asthma did not feel severe. Forty-one percent of respondents only used their inhaled medicine because their doctor recommended that they do so.

Respondents were asked whether their current treatment helped them to be free of symptoms. Sixty-two percent of the respondents felt free of symptoms with their current treatment, whereas 22% of the respondents still experienced asthmatic symptoms.

Respondents who still experienced symptoms were asked why they thought they still had symptoms. The reasons were due to various factors. Among these, respondents explained that they often tended to forget their preventive asthma medicine or were not well treated pharmacologically.

After providing the respondents with information about the new MART approach, the respondents were asked about their willingness to shift treatments. The results showed that 64% of the respondents found it likely or very likely that they would shift treatments, while 25% found it unlikely or very unlikely that they would do so.

To decide whether to switch to the new MART approach, respondents found it most important that the new treatment was more effective than their current treatment (69%), that their doctor recommended the new combination treatment (63%), and that they had extensive knowledge of the new treatment (35%). The results from the questionnaire are presented below in Figure 2.

#### 3.1.2. Support When Switching to the New MART Approach

To acquire more knowledge about the new asthma treatment, respondents would primarily seek the necessary information from the GP (74%), seek information online (68%) and ask at the local community pharmacy (47%).

To decide whether to shift to the combination treatment, the respondents primarily needed information on side effects of the combination treatment (73%), the mechanism of the pharmacological effect (67%) and general experience with the combination treatment (46%). Furthermore, the respondents wanted information on how often the inhaler should be used (37%) and whether the treatment was recommended by their GP (34%).

The three most selected initiatives that could be used to support the respondents during a change in asthma medication were consultation with their GP (85%), being handed out leaflets with information about the combination treatment (35%) and a consultation about the new combination treatment at the pharmacy (34%). Other initiatives related to information included informational videos (24%), asthma education (20%), informational apps on smartphones (14%) and other (5%), while 8% indicated no need for information.

### 3.2. Results of the Qualitative Semi-Structured Individal Interview

Nine interviews were conducted in the period from 9 May 2022 to 24 May 2022. The interviews lasted between 13–22 min (18.7 min, on average). An overview of the interviewees is presented below in Table 2.

Three themes and seven subthemes were derived from the analysis. Interview themes, subthemes and examples of quotes are presented below in Table 3. The coding process of the interview analysis adhered to the Consolidated Criteria for Reporting Qualitative Research (COREQ) [19].

#### 3.2.1. Barriers to a Shift in Asthma Treatment

##### Effects and Side Effects

One of the barriers to switching to the new MART approach that worried the interviewees the most was the effect of the treatment. Seven out of nine interviewees expressed that they were concerned about whether the new MART approach would be as effective as their current treatment, especially its effect during an asthma attack. Some interviewees did not find it relevant to shift asthma treatment, as their current treatment worked well.

Another concern was if they would get enough ICS with the new MART approach, which was not to be used daily, but only as needed when treating stage 1 and 2 asthma. In addition, the asthma patients worried about whether the combination treatment would trigger more asthma attacks, since it must be used only as needed for stages 1 and 2.

Most of the interviewees had concerns about the side effects of the new treatment. However, most interviewees believed that the advantages of the combination treatment outweighed this disadvantage.

##### The GP

It was recognized that many interviewees would like to switch to the new treatment, as they considered the combination treatment to have many advantages. However, six out of nine interviewees explained that they considered the GP to be a barrier to a possible switch to the new treatment. One of the barriers was that the process of changing their asthma treatment was time-consuming and difficult. According to four of the interviewees, it took too long before they could get an appointment with their GP, and they thought the whole process was too unmanageable and difficult. Furthermore, some interviewees pointed out that their GP was not open to their views and thoughts and would not take their suggestions into account, as the GP might have other preferences. Therefore, they could not bear to take up the fight with their GP, and they avoided expressing their wishes.

Four out of nine interviewees had a lot of trust in their GP, and they did not want to shift their treatment if their GP did not recommend the new treatment approach. Therefore, it was a barrier for the patients if the GP did not take up the conversation about switching medication. In addition, there was a tendency for the interviewees to doubt the efficacy and reliability of the new treatment guidelines because the GP had not yet recommended the new MART approach to them.

##### Lack of Knowledge

Another barrier was that some of the interviewees felt that they did not have enough knowledge about the new treatment guidelines and therefore felt insecure about changing their asthma treatment. They explained that they needed more information to be able to make a choice to switch to the combination treatment.

##### Agreement on Treatment Plan

Common to six out of seven interviewees who were being treated with two inhalers was that they often forgot their maintenance medicine, adjusted the dose themselves or opted out of it deliberately, and therefore did not use it as prescribed. Most of the interviewees could not see the reason for being treated with ICS, as they could not feel any effects from them and they did not think that they worked. Six of the nine interviewees also thought that it was impractical to take maintenance medication daily and felt that it was a limitation in their everyday life. One of the interviewees mentioned that she was worried about whether she was getting too much steroids and doubted whether it influenced her asthma treatment. Therefore, she had chosen to reduce the dose of her maintenance medicine without talking to the GP first, as she did not think she needed ICS.

#### 3.2.2. Facilitators of a Shift in Asthma Treatment

##### Trust in the GP

Another facilitator was their GP’s recommendation of the new MART approach. Four out of nine interviewees had a lot of trust in their GP making the right choices for them. One of the interviewees (IP 7) had left all decision-making about his illness and treatment to his GP. He had no knowledge of his illness and treatment, nor did he have any attitude toward changing treatment. According to him, the doctor knew what was best for him, and he would, therefore, only do what his doctor recommended.

##### Easier use with the new MART approach

According to the interviewees, the largest facilitator for switching to the MART approach was that it seemed easier and much more manageable. A common facilitator for all interviewees was that the new approach contained maintenance and reliever medication in a single inhaler. All interviewees agreed that it would make their treatment easier and more manageable if they only had to keep track of a single inhaler instead of two inhalers. Several interviewees explained that having to remember their ICS firmly limits their everyday life. Therefore, interviewees who were being treated with ICS and SABA saw the combination treatment as a great advantage, as they would no longer have to be treated with ICS daily but only use it as needed.

Furthermore, the interviewees believed that the combination treatment was an advantage, as they would also receive ICS in the case of an asthma attack. The interviewees considered this to be an advantage, as they most often tended to forget or consciously opt out of their ICS because they could not feel an effect from it.

Interviewees who were in monotherapy with SABA saw it as an advantage that they could continue to use one inhaler but also receive a more effective treatment that would treat the inflammation in the airways and not just relieve their symptoms.

#### 3.2.3. Supportive Initiatives

The results of the interviews indicated that seven out of nine asthma patients believed that a consultation with the GP to be a good initiative when deciding to switch to the MART approach. At a consultation with their GP, the asthma patients wanted information about how the combination treatment worked, the side effects, and whether it was more effective compared to their current treatment. The interviewees indicated that this would make them feel more secure during the transition process.

Another initiative suggested by the interviewees was professional advice from the pharmacy staff at the local community pharmacy. However, the asthma patients only saw this as an advantage if they did not get enough information from their own GP to start with. An intervention where the GP recommended the new approach together with the pharmacy could be a good initiative that could support asthma patients in the transition process to the new approach.

None of the interviewees felt involved by their GP in the decision-making process about their asthma treatment, and almost all interviewees wanted to be more involved in their medication. The interviewees believed that increased shared decision-making could result in them becoming more engaged in their pharmacological asthma treatment and therefore receiving better treatment. The interviewees wanted the GPs to become better at listening to the patients’ personal experiences and find treatment options that match their lives and wishes and therefore adapt to the needs of the individual.

Most of the interviewees thought that distributing a leaflet could be a good initiative to obtain concise information about the combination treatment. Another initiative that the interviewees thought could support them in a possible switch to the MART approach was online information and informational videos on the internet, as they felt this was more accessible and an illustration would be clearer and easier to remember.

## 4. Discussion

This study uniquely identified factors that may influence successful treatment shifts in asthma patients. This knowledge may be instrumental in understanding similar situations in other pharmacological settings. It must be emphasized that this study is based on 141 questionnaire and 9 interview responses and collected in a Danish setting. It is not our intention to generalize from this single case study, but rather to describe factors that may be crucial for a successful, i.e., adherent, shift to a new asthma treatment approach from a patient’s point of view.

As the inclusion of asthma patients was based on their own information on asthma diagnosis and treatment as well as the authors’ qualification of this information, it cannot be excluded that non-asthmatic patients or patients with severe uncontrolled asthma may have been wrongly included in the present study.

According to GINA 2022, the new MART approach reduces the number of exacerbations [2]. However, the present study design does not include an investigation of the potential clinical effects of a shift from a conventional asthma approach to the new MART approach. Consequently, an exploration of exacerbations was not included. Yet, it is an important factor to bear in mind in the clinical setting when choosing whether to change pharmacological asthma treatment. To recruit participants for the study, asthma patients were invited to participate either via Facebook or by physical attendance at a local community pharmacy. The results revealed that the vast majority of participants were recruited via Facebook, indicating that recruitment by random attendance of patients at a community pharmacy was not a suitable recruitment method for this study. The inclusion process may have resulted in a distortion among the respondents since only a selected segment of asthma patients who use social media was addressed. A response bias was seen among the participants of the questionnaire study, where 77% of respondents were women, and most respondents were in the younger age groups, i.e., 18–50 years. Facebook was the most popular social media platform among Danish internet users in 2018, regardless of age [21]. Nonetheless, some generations seemed to be keener on using the platform; specifically, 82% of Danish Facebook users in 2018 were aged between 16 and 24 years, whereas only 12% of the Danish Facebook users had an age of 75 to 89 [21]. Furthermore, Facebook was more common among Danish women than men that year [21]. This correlates with the present questionnaire respondent distribution based on gender and age. Thus, the recruitment strategy may have influenced the results, and future studies should seek to optimize the gender equilibrium. However, in the interview part of the study, the participants were more equally distributed with respect to gender, and the results may still be illustrative for the present case.

Below, the factors influencing treatment shift considerations and success are described.

### 4.1. Barriers and Facilitators

The results obtained from the interview showed that most interviewees were concerned about whether the MART approach would be as effective as their current therapy, especially in terms of its effectiveness during an asthma attack compared to SABA. This correlates with the results obtained from the questionnaire, where the effect of the new approach is the factor that has the greatest influence when deciding whether to change asthma treatment. An interview study by Blakeston et al. pointed out that many asthma patients have a strong attachment to SABA due to its effectiveness and immediate symptom relief during asthma attacks [22]. In addition, the study emphasized that SABA has become an emotional support for many asthma patients, especially among asthma patients who have used SABA since childhood or patients who have previously experienced severe asthma flare-ups [22]. The results from the present interviews showed that asthma patients who have used SABA for several years felt insecure when considering switching to the new MART approach. Hence, efficacy and safety may be considered potential barriers when transitioning from a known treatment approach to a new one.

Other prominent barriers in relation to the new MART approach were concerns about over- or undertreatment as well as side effects. Several asthma patients expressed that they found it unnecessary to be treated with ICS, as they did not experience the same effect as with SABA, while increasing the risk of side effects. The asthma patients, therefore, showed uncertainty about switching to the new MART approach as they were afraid of getting too much ICS. This was also the case in a qualitative study by Foster et al. exploring asthma patients’ attitudes toward using an as-needed budesonide–formoterol inhaler, where some participants articulated uncertainty regarding the necessity of adding ICS [23]. A systematic literature review by Miles et al. also pointed out that many asthma patients had concerns about the side effects of ICS [24]. It can be argued that this concern about side effects was not solely due to a switch to a new treatment approach, but probably more due to asthma patients’ general fear of side effects from ICS. Conversely, other asthma patients were afraid of not getting enough ICS with the ‘as needed’ use of the new MART approach compared with taking their asthma medication regularly daily. The asthma patients, therefore, expressed concern about whether they would experience more asthma attacks and flare-ups than before. A review by Miles et al. exploring barriers and facilitators of effective self-management in asthma found that some patients did not act on their asthma symptoms because they were not confident about recognizing their symptoms [24]. Overall, 27% of the respondents from the present questionnaire replied that it was difficult to use their asthma medication as needed, as they found it difficult to assess when they were experiencing an asthma symptom. This may prove to be the biggest challenge in implementing the new MART approach, as it requires not only asthma patients to be good at recognizing their asthma symptoms but also to be confident enough to respond to their symptoms and use their medication, which calls for supportive initiatives (see Section 4.2).

In addition to the abovementioned barriers, the asthma patients reported that it was difficult to remember their asthma inhaler, indicating difficulties with self-medication and adhering to a healthcare treatment plan. This was consistent with a systematic review investigating barriers and facilitators to asthma self-management, which showed that a lack of knowledge of asthma treatment correlated with poorer adherence [25]. Therefore, initiatives improving compliance could possibly also improve the success of treatment changes.

According to the asthma patients in this study, the largest facilitator for switching to the new MART approach was that it seemed easier and more manageable, as the same inhaler contained both maintenance and reliever medication. Similar results have been shown in a randomized clinical trial by Baggot et al., where asthma patients tended to prefer as-needed corticosteroid–formoterol therapy if they had experienced it [26]. Participants randomized to as-needed therapy were more likely to prefer not to take an inhaler every day. They also expressed a preference for being able to adjust dosing and for all asthma medications to be combined into a single inhaler [26]. A review by O’Byrne et al. suggested a reason for this patient preference. They highlight that an as-needed approach to asthma treatment recognizes that the patients have autonomy to self-adjust their dose according to their need, and their perception of need and disease control is therefore accepted [27].

A final factor influencing considerations about a shift in treatment was the asthma patients’ relationship to their GP. The asthma patients from this study expressed trust in their GP to know what was best for them. This also meant that many patients would not like to change to the new MART approach if their GP had not recommended the treatment for them. However, the interviewees also pointed out that it could be difficult to get an appointment with their GP. This combination of difficulty in acquiring a consultation with the GP and a wish to have consent from the GP presents a challenge that must be handled before a successful treatment shift can be ensured. This partnership between patients and health care professionals is a known factor influencing adherence in asthma patients [13,24,25,28], but, to our knowledge, this is the first time that this factor has also been reported to influence treatment shift considerations.

### 4.2. Supportive Initiatives

The results obtained from the questionnaire and the interview showed that most asthma patients wanted a consultation with their GP to be able to decide to shift to the new MART approach. During the interviews, it was found that none of the interviewees had experienced involvement in decision-making. According to the interviewees, involving the patient would result in them being more engaged in their asthma treatment and gaining a better understanding of their drug treatment and disease. Increased patient engagement is associated with increased patient satisfaction, increased adherence and compliance and better quality of life [29,30,31]. Therefore, an increased focus on improving patient involvement during consultations with GPs is a key initiative that could influence successful treatment shifts in asthma patients.

Another significant result from the present study was the asthma patient’s requirement to be good at recognizing their asthma symptoms and confident enough to respond to their symptoms and use their medication. Similar findings are presented in a systematic literature review by Miles et al. from 2017 exploring barriers and facilitators of effective self-management in asthma. The study found that some patients did not act on their asthma symptoms because they were not confident about recognizing their symptoms [24]. A newer literature review by Van de Hei et al. from 2021 suggested effective interventions for enhancing adherence in asthma patients to be educational and multiple component interventions, including drug reminders, inhalation instructions and counselling [32]. Consequently, it could be suggested to utilize educational interventions focusing on enhancing patient knowledge about the new MART approach as a supportive initiative for asthma patients when deciding whether to change pharmacological asthma treatments.

Most asthma patients from this study would seek information about their pharmacological asthma treatment from their GP or, alternatively, ask at their local community pharmacy to decide whether they would switch to the new MART approach. Furthermore, the asthma patients thought that distributing an information leaflet could be a good effort to obtain concise information about the new approach. Studies have previously proposed patient education, patient information, and better patient and health care professional dialogue as means to improve drug adherence [12,13,24,32]. Thus, it suggested that a patient-oriented information leaflet about the new MART approach could be developed as a patient-directed intervention and function as a supportive initiative for patients deciding on changing their pharmacological asthma treatment.

## 5. Conclusions

In conclusion, this study uniquely identified factors influencing successful treatment shifts in asthma patients that may be instrumental in understanding similar situations in other pharmacological settings, i.e., patients suffering from other chronic diseases. The results showed that factors that may influence successful treatment shifts in asthma patients were pharmacological effects, side effects, the practical use of inhalers, relationship with the GP, and patient information and involvement.

Furthermore, this study suggested that an increased focus on improving patient involvement during consultations with GPs was a key initiative that could influence successful treatment shifts in asthma patients. It also suggested that educational interventions could improve the knowledge of the new MART approach, leading to increased treatment shift success. For example, utilizing a written patient-oriented information leaflet could improve health care professional and patient dialogue, both at a general practice and at local community pharmacies.

## Figures and Tables

**Figure 1 ijerph-20-03453-f001:**
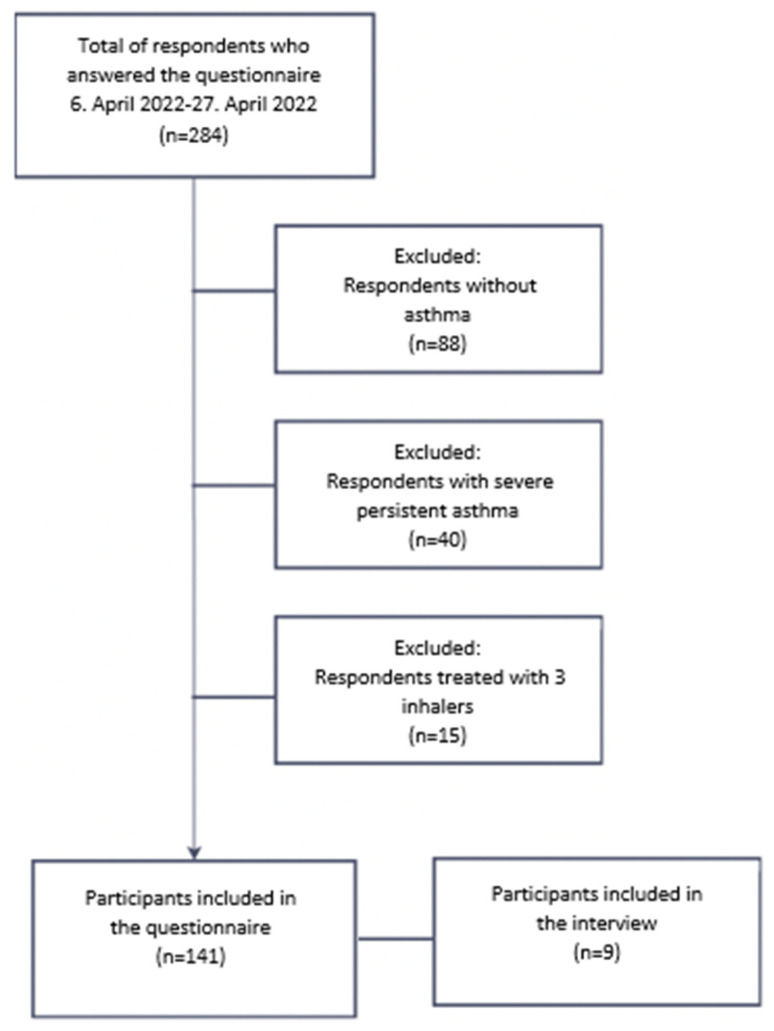
The inclusion and exclusion process is outlined in a flow diagram.

**Figure 2 ijerph-20-03453-f002:**
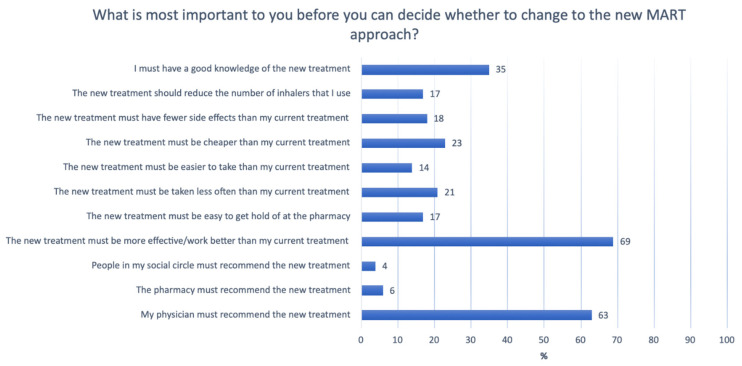
Asthma patients’ assessment of the importance of factors influencing their treatment decisions (n = 141).

**Table 1 ijerph-20-03453-t001:** An overview of the questionnaire respondents (n = 141). *Other: means here the use of ICS or LABA in monotherapy, or SABA in combination with LABA or a short-acting anticholinergic (SAMA).

	n (%)		n (%)
Gender		Occupation	
Male	32 (23%)	Student	26 (18%)
Female	109 (77%)	Self-employed	8 (6%)
Age		Employed	84 (60%)
18–30 years	62 (44%)	Retired	9 (6%)
31–50 years	56 (40%)	On leave	8 (6%)
51–70 years	19 (13%)	Not in employment	6 (4%)
71 years and older	4 (3%)	Asthma treatment	
Level of education		SABA	31 (22%)
Primary school	6 (4%)	ICS and SABA	36 (26%)
High school or other secondary education	18 (13%)	ICS + LABA comb.	27 (19%)
Vocational education	65 (46%)	ICS + LABA comb. and SABA	25 (18%)
Long higher education	52 (37%)	Other *	22 (16%)

**Table 2 ijerph-20-03453-t002:** An overview of the characteristics of the interviewees.

Interview Person (IP)	Gender	Current Inhaled Asthma Medicine	Asthma Severity	Duration of Interview (min)
1	Male	Terbutalin (SABA)	Mild	21.57
2	Male	Budesonid (ICS) and Salbutamol (SABA)	Moderate	21.11
3	Female	Formoterol+Budesonid (LABA+ICS) and Terbutalin (SABA)	Moderate	17.24
4	Male	Salbutamol (SABA)	Mild	19.35
5	Female	Budesonid (ICS) and Terbutalin (SABA)	Moderate	19.46
6	Female	Budesonid (ICS) and Salbutamol (SABA)	Moderate	18.20
7	Male	Budesonid (ICS) and Terbutalin (SABA)	Moderate	13.23
8	Female	Budesonid (ICS) and Salbutamol (SABA)	Moderate	19.18
9	Female	Budesonid (ICS) and Terbutalin (SABA)	Moderate	17.08

**Table 3 ijerph-20-03453-t003:** Interview themes, subthemes and examples of quotes.

Themes	Subthemes	Quotes
Barriers	Effects and side effects	*“Is this actually just as good when I don’t have to use it daily anymore and it’s only as needed? Am I getting what I need?” (IP9)*
	The General Practitioner	*“I simply can’t bear to go to the doctor and talk to them and they can’t really figure out what’s wrong with me.” (IP4)*
	Lack of knowledge	*“I don’t feel like you get enough information.” (IP3)*
	Agreement on treatment plan	*“I haven’t talked to my doctor about it, but I’m trying to taper off the preventive [medication] because I don’t need it.” (IP5)*
Facilitators	Trust in the General Practitioner	*“Yes, I have to get my doctor’s approval for that [treatment change]. I think doctors know best.” (IP5)*
	Inhaler use	*“It will make it easier, because then I will take it [the medication]. Because when I take two [inhalers] I forget one of them, sometimes.” (IP4)*
Supportive initiatives	Oral consultation	*“So probably more information from the doctor when you get your prescription for the first time. Or maybe the pharmacy could give a quick introduction: Do you know how to use this?” (IP3)*
	Written information	*“I once got a pamphlet from the pharmacy, maybe another [pamphlet] like that? Just to get some general knowledge. Yes, that would be great. Definitely!” (IP9)*

## Data Availability

The project is registered at the Capital Region of Denmark’s record of processing activities according to the European Union General Data Protection Regulation (GDPR) art. 30 (ref. P-2023-45). Research based on quantitative questionnaires and qualitative semi-structured interviews are exempted from a requirement for ethics approval according to Danish Law governing ethical approvals of health science research projects (LBKG 2017 1083) § 14 (2).

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
