# Peer review of "A Shift in Asthma Treatment According to New Guidelines: An Evaluation of Asthma Patients’ Attitudes towards Treatment Change"

_ijerph, 2023, doi:10.3390/ijerph20043453_

Round 1
Reviewer 1 Report
General comments
The GINA 2022 guidelines recommend to apply a “MART” strategy for the treatment of mild and moderate asthmatics. “MART” is based on a treatment of ICS-formoterol for both “Maintenance And Reliever Therapies”.
Based on responses to a virtual questionnaire among 141 asthmatics and 9 interviews among 9 other asthmatics, the authors suggested that applying the new GINA treatment recommendations are limited by the perception of the patients of pharmacological effects, side effects, the practical use of inhalers, relationship with the GP, and patient information and involvement.
It is a descriptive study of the different responses that are sumarized and subjectively interpreted by the authors, all linked to a pharmacology Department. None of the author is a clinician, pulmonologist or GP or other.
There is no statistical evaluation of the responses and this is a major concern that preclude the final conclusions.
Specific comments
1/ the authors claims that GINA 2022 recommends that all asthma patients, regardless of their severity, receive ICS in combination with a LABA as both maintenance and reliever therapy (MART) and that this LABA should be formoterol; for the the authors, GINA does not recommend the short-acting β2-agonists (SABAs) in monotherapy.
In fact GINA 2022 recommends 2 approaches to treat mild and moderate adult asthmatics:
“track 1” : Step 3 treatment based on low dose-ICS formoterol as maintenance and reliever therapy
“track 2”: maintenance ICS-LABA plus as-needed SABA
(see GINA guidelines 2022, page 56 , Box 3-4, Bi)
2/ before considering any treatment, GINA 2022 recommends to “first confirm that the symptoms are due to asthma and identify and address common problems such as inhaler technique, adherence, allergen exposure and multiple morbidity”.
In the present study the diagnosis of asthma was based on the patient personal knowledge and not on the presence of combined symptoms with an objective measurement of airways obstruction variability (post- bronchodilator spirometry and/or bronchial hyper-reactivity).
I can hypothesize that a proportion of the included subjects are not asthmatics.
The resent study has not evaluated the common problems related to asthma, before to discuss with the patient the MART strategy.
3/ how can you explain the gender disequilibrium (M/F = 32/109)?
4/ patients not controlled by LABA-ICS + SABA (i.e. 18% of your population) should be defined as severe asthma (and not recommended to switch to MART strategy)
5/ in table 1, 22 (16%) of the patients are treated with “other”. What mean “other”?
6/ the major interest of the “MART” strategy is to prevent the exacerbations of asthma. Unfortunately there is no data about the history of exacerbations about the 141 subjects.
Author Response
Many thanks for the good and relevant comments on our manuscript. We are happy for the opportunity to provide clarifying answers.
General comments
As a reply to the comment about author profession:
The authors are clinical pharmacists whose curriculum is in close relation with clinical pharmacologists. The authors work in the health care system of Denmark in a clinical setting. The authors work in constant collaboration with both hospital specialists, e.g. pulmonologists, and general practitioners. Furthermore, the last author has a Ph.D. in pharmacology.
As a reply to the comment about the lack of statistical analysis:
The study design and method description adhered to the JBI Critical Appraisal Checklist for Qualitative Research shown in Appendix 2 [1] and the Consolidated Criteria for Reporting Qualitative Research (COREQ). [2] In addition, the case study analysis was performed by comparing the present results to facts in published literature according to case study methods. [3]
Specific comments
As a reply to comment 1:
We thank the reviewer for the perceptual comment about the GINA 2022 guidelines recommendations and we have corrected the manuscript text accordingly (p. 2).
As a reply to comment 2 about asthma diagnosis:
The comment has been noted and the manuscript has been corrected accordingly by adding a description of how the diagnosis of asthma has been estimated (p. 3) and a discussion of this limitation in the discussion (p. 11).
As a reply to comment 3 about gender disequilibrium:
The comment has been noted and the manuscript has been corrected accordingly by adding a discussion of factors influencing patient responses (p. 10).
As a reply to comment 4 about the definition of severe asthma:
The comment has been noted and the manuscript has been corrected accordingly by adding an elaboration on the study design and inclusion criteria (p. 3) and by stressing, that asthma patients only were recommended to change to the new MART approach if they had controlled and stable asthma (p. 5).
As a reply to comment 5 on Table 1:
The comment has been noted and the manuscript has been corrected accordingly by adding a description of the term ‘other’ in Table 1 (p. 5).
As a reply to comment 6 about history of asthma exacerbations:
The comment has been noted and the manuscript has been corrected accordingly (p. 11).
The present qualitative study investigates asthma patients’ attitudes towards a possible treatment change. The study design does not include an investigation of the potential clinical effects of such a shift. Consequently, an exploration of exacerbations was not included. Yet, it is an important factor to bear in mind in the clinical setting when choosing whether to change pharmacological asthma treatment.
References:
[1] Lockwood C, Munn Z, Porritt K. Qualitative research synthesis: methodological guidance for systematic reviewers utilizing meta-aggregation [Internet]. Int J Evid Based Health; 2015. Available from: https://view.officeapps.live.com/op/view.aspx?src=https%3A%2F%2Fjbi.global%2Fsites%2Fdefault%2Ffiles%2F2021-10%2FChecklist_for_Qualitative_Research.docx&wdOrigin=BROWSELINK
[2] Tong A, Sainsbury P, Craig J. Consolidated criteria for reporting qualitative research (COREQ): a 32-item checklist for interviews and focus groups. Int J Qual Health Care. 2007 Dec;19(6):349–57. [3] Gustafsson, Johanna J. Single case studies vs. multiple case studies: A comparative study. 2017:15.

Reviewer 2 Report
Thank you for your paper.
Please add information about the ethical approval for your study.
Given the way your research was conceived we can not speak about "influencing" factors. The statistical analysis would need to support this.
The introduction part needs to be completed with relevant international studies on this topic.
The findings can not be extrapolated, for instance the situation related to GP, as it depends on the structure of the medical system in each country.
Author Response
Many thanks for the good and relevant comments on our manuscript. We are happy for the opportunity to provide clarifying answers.
As a reply to comment about ethical approval:
We thank the reviewer for the perceptual comment about the missing information about our ethical approval and we have corrected the manuscript text accordingly (p. 13). The project is registered at the Capital Region of Denmark’s record of processing activities according to the European Union General Data Protection Regulation (GDPR) art. 30 (ref. P-2023-45). Research based on quantitative questionnaires and a qualitative semi-structured interviews is exempted for ethics approval according to Danish Law governing ethical approvals of health science research projects (LBKG 2017 1083) § 14 (2).
As a replay to comment about “influencing” factors:
The comment has been noted and the manuscript has been corrected accordingly by changing the wording in the discussion, conclusion and abstract (p. 1, 10 and 12).
As a reply to comment about the introduction:
The comment has been noted and the manuscript has been corrected accordingly by adding new literature (p. 1-2).
As a reply to comment about transferability:
The comment has been noted and the manuscript has been corrected accordingly by rephrasing the discussion, conclusion and abstract (p. 1, 10 and 12).

Reviewer 3 Report
Congratulations nice idea, and well written manuscript. I would add a table containing an overall response to the result section and check for factors influencing patient responses such as: age, sex or education.
Also the definition on how the diagnosis of asthma has been given should be provided.
I would appreciate larger study group however i do understand that this is not doable now.
Author Response
Many thanks for the good and relevant comments on our manuscript. We are happy for the opportunity to provide clarifying answers.
As a reply to comment about additional table:
We thank the reviewer for the comment about adding the table containing an overall response to the result section. We already have a figure containing an overview of the overall number of participants, but we have corrected the manuscript text by updating Figure 1 to contain both information about questionnaire respondents and interview participants (p. 4).
As a replay to comment about influencing factors:
The comment has been noted and the manuscript has been corrected accordingly by adding a discussion of factors influencing patient responses (p. 10).
As a replay to comment about a definition of asthma diagnosis:
The comment has been noted and the manuscript has been corrected accordingly by adding a description of how the diagnosis of asthma has been given (p. 3).
As a replay to comment about study size:
The comment has been noted and the manuscript has been corrected accordingly by adding a emphasis in the discussion on study size and generalizability (p. 10).

Round 2
Reviewer 2 Report
no further comments